# How Can Physical Inactivity in Girls Be Explained? A Socioecological Study in Public, Subsidized, and Private Schools

**DOI:** 10.3390/ijerph19159304

**Published:** 2022-07-29

**Authors:** Rodrigo Soto-Lagos, Carolina Cortes-Varas, Solange Freire-Arancibia, María-Alejandra Energici, Brent McDonald

**Affiliations:** 1Faculty of Education and Social Sciences, Andres Bello University, Viña del Mar 2520000, Chile; a.cortesvaras@gmail.com (C.C.-V.); solangefreirearancibia@gmail.com (S.F.-A.); 2Faculty of Psychology, Alberto Hurtado University, Santiago 9160000, Chile; menergic@uahurtado.cl; 3Institute for Health and Sport, Victoria University, Melbourne, VIC 3011, Australia; brent.mcdonald@vu.edu.au

**Keywords:** physical inactivity, gender, physical education, socioecological model, physical activities, girls, public policies

## Abstract

In the last few years, the World Health Organization has highlighted that physical inactivity is a global issue affecting women to a greater extent than men. Faced with this, different nation states have developed public policies to reduce physical inactivity at school; however, the biomedical and individualistic models used have generated widespread criticism, as figures remain the same. In the context of failed interventions on increasing levels of physical activity, this study utilizes a socioecological model to analyze and understand how physical inactivity is reproduced in girls in the Chilean education system. A qualitative study was implemented, as it allows a focus on the entailed meanings, context, and processes. Active semi-structured interviews were conducted with 40 groups comprising headmasters, teachers, non-teachers, students, and families. The results show that physical inactivity is linked to factors that go beyond the individualistic model; that is, consideration must include intrapersonal, interpersonal, organizational, community, and public policy dimensions. Furthermore, gender stereotypes gain relevance in physical education classes, in addition to friendships and family, teaching, and administrative work, access to safe play areas, use of spaces, and widespread cultural factors associated with men and women. This study concludes that the assessed gender differences should be approached from a pedagogical perspective beyond common sense, further reporting that the individualized explanation for physical inactivity is irrelevant to answer why women are more inactive than men.

## 1. Introduction

Physical inactivity has been cited as the 4th leading cause of death worldwide [1], and in recent years, the World Health Organization (WHO) reported that this global issue is more frequent and severe in girls and women [2,3,4,5]. The WHO further stated that schools should promote physical practices, such as exercise and sports, to ensure that recommendations for daily physical activity are adopted and achieved in student populations [6].

Chile is no exception in relation to the population achieving the required levels of physical activity encouraged by public health guidelines. The 2017 National Health Survey [7] reported that 86.7% of people do not exercise at least 3 times per week and noted the worrying trend that 9 out of 10 women, and people from lower socioeconomic backgrounds, were deemed sedentary. Further, 96% of children with less than 8 years of schooling were also considered inactive.

The Ministry of Education generates actions to decrease physical inactivity at school. The National School and Scholarship Assistance Council (JUNAEB, for its Spanish acronym) presented the nutritional map of the country’s educational institutions, indicating that 50% of students attending pre-kindergarten to 9th grade were overweight or obese, as a result of physical inactivity, among other causes [8]. To overcome this situation, JUNAEB promotes the programs “Escuelas Saludables para el Aprendizaje” (Healthy Schools for Learning) and “Contrapeso” (Counterweight), which are aimed at reducing physical inactivity and obesity. Meanwhile, the Ministry itself published the Guidelines and Considerations for School Physical Activity during the pandemic [9]. In December 2021, the Ministry of Sports declared that 78.5% of children aged 11–17 were physically inactive and highlighted that girls are the most affected group (84.9%) compared with boys (71.8%) [10].

In line with this problem, from 2014 onwards, all Chilean schools began to be evaluated through the education quality measurement system (SIMCE, Spanish abbreviation) regarding the promotion of healthy lifestyle habits in the school community [11]. Moreover, in 2015, the Ministry of Sports created the “Escuelas Deportivas Integrales” program (Comprehensive Sports Schools), which offered an increase in physical activity in public schools alongside professional mentoring by psychologists and nutritionists. Under Sebastián Piñera’s government (2018–2022), this program changed its name to “Crecer En Movimiento” (Growing on the Move), maintaining the same intervention strategy.

Today, different schools across the country implement and/or create interventions aimed at reducing the population’s physical inactivity. Therefore, educational institutions have become privileged spaces to affect behavioral changes in the physical activity of students [2,6]. In this context, a bill requiring Chilean schools to undertake 15 min of daily physical activity was passed (see: Boletín 11518-11), an initiative that has led to tension in the school communities, as not all of them have the resources to adopt this bill.

As for the public and private initiatives that have attempted to reduce physical inactivity, several studies [11,12,13,14,15] have shown that interventions focused on prompting individual behavioral changes concerning diet and amount of physical activity do not lead to significant changes.

Some explanations have noted that the discourse inspiring the development of healthcare policies is based on medicine and biomedicine [16], and that the actions suggested by them have focused on individuals [17]. This has been consistent with linear logic with respect to problems and solutions, as the idea that people should be responsible and make the correct choices to reduce physical inactivity in their lives has been established, which several authors [18,19] have called the moralization of everyday life, which ignores the various social determinants of health [1,12].

In agreement with the above, different studies have aimed to contribute to the project for the reduction of physical inactivity across the world. In this sense, Mielke et al. [20] highlight the fact that, except for only 8 of the 142 countries studied, women were more inactive than men, a situation that aligns with the Chilean figures.

As for the approach adopted to reduce physical inactivity, Westerbeek and Eime [21] suggest that a transition has been observed in the last few years regarding public policy models—from a sports-competition to a sports-recreation model that is fully integrated into individual’s daily lives. The sports-recreation model is known as a comprehensive sports ecosystem because, and similar to Muzenda et al.’s [22] proposition, it considers trans-sectoral cooperation, and the incorporation of contributions from other sectors such as health, education, transportation, and urban planning to be essential.

Recent studies show that including children’s preferences is crucial for them to acquire the habit of exercising. In other words, to achieve positive changes in public policies, children should be included in this process by listening to their opinions, and regarding them as active subjects capable of engaging in reality [23].

Considering the pandemic, several studies [24,25] mention that physical inactivity and sedentary behavior in children and youth increase as a result of COVID-19. This last study highlights that the gap between men and women remained stable during this period and that keeping schools open facilitates physical activity, which decreased when schools were closed. This suggests the relevance of school establishments regarding the promotion of physical activity in children and youth.

### 1.1. Physical Inactivity in Girls

Isorna et al. [26] indicated that the physical (in)activity of women is influenced by gender stereotypes that condition the pedagogical processes and students’ aptitude. These processes affect personal interests and motivations, thus influencing their level of involvement in some physical activities.

Castejón and Gimenez [27] revealed that in physical education (PE) class, boys preferred sports, while girls opted for expressive activities, such as dancing. According to Alsarve [28], this could be associated with the masculinized approach used in these classes, as it highlights that the beliefs and views of masculinity and femininity are stereotyped. Based on this work, masculinity was linked to traits such as physical strength, roughness, and agility; conversely, however, feminine roles were related to household chores, fragility, rhythm, subordination, and weakness.

According to Alvariñas and Pazos [29], Curieses [30], and Mujica [31], these stereotypes threaten women’s involvement in traditionally male-dominated disciplines and are justified by the idea of sports as an impractical and dangerous activity for women. Similarly, imagery has been constructed of women who engage in sports (either recreationally or competitively) as transgressing gender norms.

As for the role played by PE, several authors [29,30] express that this discipline is permeated by male and female stereotypes that are present at school, validating the differences between genders and thus affecting school culture. In this sense, Cameron and Humbert [32] indicated that this situation is closely related to the roles played by men and women in society, concretely representing this through the existing differentiation between sports designated as “male” or “female.” In this way, gendered differences in PE and incidental physical activity (play) at school become naturalized to educators [33].

The above, according to Parri and Ceciliani [34], would translate into differential behaviors by teachers, conveyed in subtle and explicit manners through various learning expectations. The authors further indicate that the differential expectations also affect girls in key aspects related to their identity and self-esteem, negatively conditioning their engagement with physical activity, and pushing them into smaller spaces, instilling submission, and a lower acceptance of physical contact. In contrast, Fissette [35] explained that female students consider differential treatment and gender and sports-associated stereotypes to be unfair, contributing to the ‘why’ they drop out of sport and physical activity.

In line with this issue, Mayorga et al. [36] suggested that unequal treatment is one of the barriers perceived by teenagers when doing physical activity, subsequently influencing their habits as adults. Moreover, Sánchez et al. [37] stated that the role played by teachers and their behavior concerning gender is conditioned by the influence of other factors, such as socioeconomic, environmental, family, or cultural variables. For this reason, teachers face the challenge of integrating these variables into their work and promoting innovations in each class for students to feel involved in the educational process, rather than feeling excluded owing to their gender, abilities, or physical qualities.

In the same vein, a recent study conducted by Energici et al. [38] reported that the abovementioned situation may be explained by the reproduction of gender stereotypes created in and through the practice of physical activities. This study may be considered a radical criticism of the individualistic model for addressing physical inactivity, as it suggests that the factors affecting this problem go beyond individuals.

The concern with children’s and adolescents’ experiences of physical activity is the relationship between these experiences and lifelong engagement with physical activity [39]. The school and PE are likely to be important locations for these experiences. If PE is gendered to create negative experiences for girls, then the likelihood of physical activity in adulthood for women may be reduced [40].

### 1.2. The Socioecological Model

Criticism of the individualistic perspective that seeks to reduce physical inactivity at school has been identified in recent years. One critique suggests that this perspective has not considered that each educational reality has its own school culture, and that the provision of a standard measure may overshadow both the local problems and the strengths of a community to solve them [6]. Similarly, studies report that including health in PE has led to activities that children are reluctant to do [41], as well as triggered emotions that may result in a preventive pathologization of their everyday life [42], causing fear [43,44] and annoyance [45]—emotions that fail to bring about the change that public policies seek [46].

These studies considered the socioecological model developed by Bronfenbrenner [47], who aimed to make the various dimensions interacting in the continuance and promotion of several social problems visible, specifying the existence of a dependency between systems and agents, which, in turn, exert a great influence on individuals’ behaviors and practices.

Specifically, research on physical inactivity from this approach includes studies conducted by other scholars [48,49,50]. McLeroy et al. [48] suggested an ecological model featuring five degrees of influence (at the intrapersonal, interpersonal, institutional, community, and public policy levels), which should be considered in any intervention promoting a reduction in physical inactivity, as proposed by the WHO. Sallis et al. [51] highlighted the environmental and political influences in four fields of active life (recreation, transportation, occupation, and home) and called for collaboration with policy researchers to improve the probability of translating research results into changes in the environments, professional practices, and public policies. Bauman et al. [52], for their part, focus on the areas of life wherein exercise is done (home, work, transportation, and leisure time) and mention that these are different according to the country, age, gender, ethnic origin, and socioeconomic status of each context.

In the last few years, different authors [53,54,55,56,57] have adopted this approach to assess physical inactivity. In a cross-sectional way, they indicate that at the intrapersonal level, gender, age, ethnic group, and self-concept are relevant aspects, while at the interpersonal level, the family environment and meeting with friends are key points to explain engagement in physical activities. As for organizations and schools, the role of teachers and principals should be emphasized, whereas at the community level, socioeconomic aspects, educational level, and screen time are factors affecting physical inactivity. From the public policy dimension, access to facilities and safe neighborhoods are crucial factors for individuals to undertake physical activity.

Taking into account the criticisms of the individualistic and biomedical approach to achieving any meaningful change in the physical activity levels of populations, especially in relation to gendered differences, our study seeks to assess the effects of the public policy agenda to increase physical activity in Chilean schools. We utilize a socioecological framework to engage with a variety of actors in schools to understand the complexities of implementing such agendas, especially in relation to the reproduction of physical inactivity in girls in educational settings.

## 2. Materials and Methods

### 2.1. Methodology

This study is based on Sparkes and Smith’s [58] postulates regarding qualitative research. The authors claim that this type of study must focus on the meanings, context, and processes that condition the issue that is the object of research. In this case, knowledge is built based on the opinions and experiences of the individuals affected by the public policies that seek to reduce physical inactivity in Chilean schools.

### 2.2. Participants

A theoretical sampling of schools was carried out with the understanding that all public, subsidized, and private establishments have been invited to carry out actions to reduce physical inactivity. In this context, based on the institutional relationship of the university sponsoring this research, three schools were invited for each type of administration that exists in Chile. The invitation was made directly to the email of each director, who invited the rest of the educational community.

The sample comprised eight schools: three public (all of primary level), two subsidized (both with primary and secondary level), and three private (one primary level and two primary and secondary levels). Five roles were identified in each one, which led to the development of a sample including principals (N = 31), teachers (N = 54), non-teachers (N = 37), students (N = 59), and families (N = 43). The participants came from the same organizations that agreed to participate. Access to the participants was through the management teams, who invited them, and they freely agreed to participate in the interviews. Likewise, the directors were asked to ensure parity between men and women in the calls.

It should be noted that the three types of schools have a director, but they are managed by a management team made by the director, vice director, curricular manager, and behavior manager. In the call for participants, the teams were invited all together because the decisions of the Chilean schools were taken in groups. The teachers were from PE and from other subsectors, as the practices to reduce physical inactivity affect the community, and all teachers can have an opinion about it. The non-teaching staff were administrative staff, support professionals (psychologists, social workers, speech therapists), and auxiliary and cleaning staff; they were included in the interviews in the understanding that practices to reduce physical inactivity affect the community and because their experience could add more layers and nuance on the topic. The students who participated, at the time of the interviews, were between seventh grade of elementary school (11–12 years old) and 4th grade of secondary school (17 years old). In families, not only fathers and mothers were invited, but also caregivers, who in some interviews were grandparents and uncles. Students and families were not always related.

In total, 40 group interviews were conducted, totaling 224 individuals participating in the study. All participants were invited by the manager team of each school, and their participation in the research was voluntary. Public establishments were located in the Metropolitan and Valparaíso region, whereas subsidized schools were situated in the region of Valparaíso and O’Higgins, and private institutes were in the Valparaíso and Magallanes areas.

To contribute to the understanding of the Chilean context in economic terms, public schools depend exclusively on the state for funding. The subsidized schools receive a monthly amount of public money for each student, and they have the option of receiving money from the families; that is, they receive public and private resources. Private schools are only maintained with the contribution of families. As for the curriculum, all schools must meet the minimum content proposed by the Ministry of Education. However, if some want to teach more content, they can do so (for example, languages, science, or other sports). Private schools are most likely to teach more content.

### 2.3. Data Production

Data were obtained from semi-structured active–reflective group interviews [59,60]. Considering group diversity, the research team asked each administrative team to manage the participants, who, in turn, voluntarily and freely consented to take part in this study.

The interviews were divided into three sections. The first delved into the work of each position within the school community. For example, tell us about your work and include all the information you consider relevant to understand what you do in this school. The second section asked about their experience with public policies aimed at reducing physical inactivity. For example, what actions do you carry out as an establishment to reduce physical inactivity? Do you have any link with a public service to reduce physical inactivity in your school? Finally, the third section sought information on the complexities observed by each role when trying to reduce physical inactivity at school, for example: How do you deal with physical inactivity in your establishment? What are the complexities experienced in increasing PA in your school? Do gender, socioeconomic level, cultural background, and educational level play any role? It is worth noting that, in view of the COVID-19 pandemic, most interviews were conducted via Zoom. Each interview lasted for around 45–60 min.

### 2.4. Analysis

To perform the analysis, the dimensions of the abovementioned socioecological model were used [61]. This proposal was divided into five aspects: the intrapersonal, interpersonal, organizational, community, and public policy levels. It is worth highlighting that considerable attention has been given to gender dimensions, as, according to what has been stated earlier, girls and women embody the most worrisome figures concerning physical inactivity.

The premises of the discourse analysis were followed to perform the analysis itself [62,63,64,65]. In line with these ideas and the socioecological model, inductive logic was adopted [66,67,68], in addition to deductive logic, from the categories provided by the model. To conduct the interpretative task, Moreno, Rivera, and Trigueros’ study was considered [69].

For the validation of the research results, transferability [70] and representativeness [71] criteria were used. Similarly, all bioethical guidelines proposed by the Bioethics Committee of the Andrés Bello University were followed.

## 3. Results

The results of this study are presented after performing the analyses, following the intrapersonal, interpersonal, organizational, community, and public policies referred to in the socioecological model.

This study allowed for the gender-based assessment of physical activities that different educational actors (principals, teachers, non-teachers, students, and family) promote to reduce physical inactivity at school.

### 3.1. Intrapersonal Level

The intrapersonal level considers the biological and psychological factors that influence an individual’s behavior. Based on the data obtained, the educational establishments under study show that physical inactivity in children is conditioned by physical and emotional changes that can be explained by their life cycles. In a cross-sectional way, all educational actors indicate that no play differences are observed in children aged 5–11 years (1st–6th grade), but boys and girls aged 11 and 12 go through a transition point regarding their engagement in this practice, which was attributed to the onset of puberty and subsequent breast development and menstruation in girls. Changes due to puberty were seen as a reason why girls disengaged with PE classes, sports, and physical activities at school owing to the emotional changes generated by these physical and social changes.

Teachers and principals attribute girls’ lack of motivation to the effects of adolescence and puberty. Although this is a common situation for most students, “not all of them” experience decreased motivation in class.


*In sixth, seventh, eighth grades (…) we usually see unmotivated girls in class, and boys always seem to be more involved, so something goes on during said maturity, adolescence, and puberty cycle, when girls lose motivation in their physical education classes. This is not the case for all of them, but in general. I have experience teaching at all levels, and we always see a decrease in the girls’ engagement in [high school].*

*(Subsidized School No. 2, Teacher)*


In the excerpt above, girls’ decreased involvement is the result of their life cycle, a situation “they undergo during their maturity process”; furthermore, the teacher expresses that no differences in the involvement between girls and boys can be observed in the first school cycle (1st–5th grade), as is the case when they reach sixth grade, and onward. The teacher indicates that “their involvement always decreases slightly in [high school],” an expression that normalizes the students’ attitudes, behaviors, and emotions, without making the social dimension of the teaching-learning process visible.

In connection with the above, another aspect that turned out to be relevant in the intrapersonal dimension was the biological differences between men and women. According to the students themselves, before adolescence, no physical differences can be seen between them, but as they grow, the characteristics that make teachers draw a distinction between the difficulty/weight of exercises for men and women emerge, as can be noted in the following excerpt.


*Student No. 2: Because of the biological differences between men and women.*

*Interviewer: I see... and what are said differences like?*

*Student No. 3: Different muscle mass; ours is lower than boys.*

*Student No. 2: As already mentioned, it varied based on our age, as you could say that there are not so many differences in terms of school development when we are young. As we reach eighth year, boys and girls begin doing different exercises.*

*Student No. 3: So, in basic education grades, boys and girls are asked to “do 10 crunches,” but in middle school, girls are told to do 30 push-ups, for example, whereas boys are asked to do 40 of them, or something like that.*

*(Private School No. 2, Students)*


In view of the differences between boys and girls, students highlight that “biological” distinctions can be identified in the PE class, associated with a “different muscle mass,” making reference to the fact that women are not as strong as men. Based on this role, they agree that when younger children perform the same activities, but after eighth grade, contrasts become more noticeable, for example, regarding demands. The expression “in middle school, they are asked to …” reveals that the requests are different for boys and girls, as the latter, for instance, are requested to do 30 push-ups, whereas boys must do 40. While this situation can seem natural to them, it can be regarded as an externally imposed limitation, given that expectations in terms of demand come from teachers who “tell you” the amount of exercise to be done.

Another relevant aspect that may account for the low involvement of girls in PE or sports classes is menstruation and breast development.


*Student No. 1: Besides, another thing that people do not talk about much… of course, women menstruate once a month so, in some classes, you cannot clearly ask a girl who is on her period to run 15 min. At least I consider it to be really uncomfortable, to run and feel you can stain your clothes. I am new; I entered the school in first grade, and I barely know my peers, so the first class together was really uncomfortable; I was on my period, and it was really difficult, not knowing my partners, and all that…*

*(Private School No. 2, Students)*


In view of the girls’ opinion, this seems to be taboo, as this is not something that people “talk about” in face-to-face classes. Being on their menstrual cycle causes discomfort in female students, affecting their participation in physical activities, as this is a subject rarely addressed. Girls are not sufficiently confident to talk about it with their peers or teachers.

As indicated in the excerpt, menstruation is an issue that generates concern in girls, which is enhanced when they are “new” students, as they are not close enough to their peers. This would lead to different emotions, and getting involved in class becomes a “difficult” and “uncomfortable” situation that requires building confidence to overcome it. This presents a situation that should be explored beyond the common sense of students and educational actors.

Based on their experience, teachers suggest that breast development is a situation associated with shame, an emotion that students experience when they feel that they are being observed or criticized during PE class:


*Teacher No. 3: Maybe, as they are going through a growth stage, they feel more ashamed if they have bigger breasts, and aspects like that; they do not want to be exposed in front of others or criticized, I think.*

*Teacher No. 1: They seem to be insecure about their physical aspect, you know? About their appearance … [they feel insecure]*

*Teacher No. 2: [And frequently, in the second course] that is really common, girls are really insecure, and in all senses, when they have to take part in activities, if they have to sing or move, there is little involvement from them.*

*(Public School No. 1, Teachers)*


Therefore, according to the data obtained, physical changes, such as breast development and menstruation, are associated with non-positive emotions in the PE class. This occurs in the three types of education administration (public, subsidized, and private), wherein emotions such as insecurity, shame, and lack of self-esteem and self-acceptance develop, evident from the concern about their body and, more specifically, about their peers’ opinions of them.


*Student: Yes, I remember that in other years, and even before COVID, girls in dressing rooms were more concerned about looking good rather than being worried about having shoes, for instance, or they did not want to exercise wearing these leggings as people would be looking at them, or… stuff like that. I believe that these effects girls’ performance because I’ve never seen differences made by the establishment or teachers*

*(Subsidized School No. 2, Student)*


The quote above shows that dynamics are developed in PE classes that, although they may concern biological aspects associated with development stages, agree with the relationships built between peers. For instance, in the statement wherein a student says that she does not want to exercise wearing leggings, as people would be looking at her, she gives a social justification that may explain the high rates of physical inactivity in women, which is related to the fear of exercising owing to the opinion that other individuals may have about their bodies.

The above statements made by teachers and principals can be explained as self-esteem problems, which may lead to insecurity. During the pandemic, self-esteem problems became evident as girls attended lessons, but their cameras were switched off.


*As for girls’ drop out, if I consider the stage, age of students, and girls who become judgmental, self-esteem is crucial to prevent this issue. Unfortunately, their self-esteem makes them insecure, and this lack of security is evident now in the light of the pandemic, when we insist on asking them to switch their cameras on, and they try to avoid showing themselves, because they fail to accept their bodies. So, this becomes a key issue when they stop doing physical activity, showing themselves, or taking care of themselves*

*(Public School No. 1, Teacher)*


The excerpt above reports that the insecurities identified in students—illustrated with expressions such as “they avoid showing themselves,” “they do not switch their cameras on,” and “they do not accept their bodies”—may be the reasons why they stop doing physical activity, showing themselves, and, thus, stop “caring about themselves.” This passage also provides a social and pedagogical explanation to address inactivity in women and in school in general.

### 3.2. Interpersonal Level

The next dimension developed within the socioecological model is interpersonal, associated with social networks (formal and informal) and social support systems, including family and friends.

From the data obtained, students make a difference between their involvement in face-to-face physical education classes and those delivered via digital platforms as a result of the COVID-19 pandemic. The first was associated with fun and positive emotions, whereas the online format was linked to being alone and a lack of interaction with peers—a non-positive perception.


*Student No. 2: Yes, face-to-face classes were funnier because we were together … we always run at the beginning, while at the end we played games, so physical education classes were really fun under said modality.*

*Interviewer: And what are classes like now?*

*Student No. 2: Now … well, the virtual class is good too, but… not as fun as it was before because you are alone, and you don’t speak to almost anybody, so… it’s okay, but face-to-face classes are better.*

*(Private School No. 3, Students)*


When pointing out that “we were together,” the student highlights meeting with peers as the reason for “fun” in the PE class. Similarly, this positive emotion is linked to the games played at the end of the class, which are positively associated with face-to-face modality and gathering with friends. For their part, virtual lessons were related to being alone and unable to talk to anyone, aspects that would decrease the motivation that the group feels over individual and lonely practice.

In the same vein, a student from another type of establishment states that if he needs help during the virtual class, he never asks for it. However, in face-to-face lessons, “he turns to a friend for help.”


*Student No. 1: Another could be when we want to get help, but do not feel confident, so we just avoid doing it in online lessons (…) In face-to-face classes, if you do not feel sure about something, you can ask a friend for help, who acts as a companion.*

*(Subsidized School No. 2, Students)*


This excerpt highlights that social support from friends in PE classes is relevant for students. A significant figure accompanying the student, such as a friend, can help students ask for help and perform activities that can be regarded as challenging.

With regard to family support, different experiences were observed in face-to-face and online lessons. Parents mentioned that virtual lessons led to less positive emotions regarding participation, and technology was identified as a challenge for families, as the link with digital devices increased with this modality. According to a guardian, sedentary behavior as a result of technology was greater in times of COVID but even higher in women than men.


*Parent No. 1: My little son (…) for example, I have a treadmill, and he turns it on by himself, starts running, jumps on the trampoline… he is always looking for something to do, because he is not the kind of child that can be entertained with a cell phone or the TV. Thank God! he is not the kind of child who keeps quiet sitting, no … he is always in the backyard, playing with the dog; he is restless. This is not the case with girls, who play with their phones, use TikTok, etc. It’s difficult for me to take my daughter away from the TV, cell phone … but yes, I can tell the difference between a girl and a boy, and it is worse when they are children of different ages.*

*(Public School No. 1, family member).*


In the above statement, a scenario is built wherein the motivation for physical exercise varies between men and women, and this could be conditioned by the development stage, as already mentioned. At this point, a teacher suggests that families play a key role in promoting physical activities and sports in boys and girls.


*I think this is a cultural matter, I don’t know ... I still believe physical activity is not promoted by families in the same way for girls or boys … as with soccer, for instance, nowadays, girls are highly motivated to play soccer, but it is not natural for us to understand that a girl may like to do a sport, like girls who like doing sports or exercising are really peculiar. The case of men is more general … those who like doing physical activity. But I do believe that both genders have reduced their willingness to exercise, and I think this can be partially explained by their sedentary behavior.*

*(Public School No. 2, Teacher)*


The expression “physical activity is not promoted by families in the same way for girls or boys” shows that men and women are encouraged to exercise in different ways. Although the number of girls doing sports is growing, the teacher highlights that “girls who like sports are peculiar,” thus reinforcing the idea that they may be less motivated than their male peers. Nonetheless, according to the teacher, “both have decreased their willingness to do physical activity,” which poses a challenge for schools and families as sedentary behaviors increase in the pandemic context.

### 3.3. Organizational Level

This dimension refers to the influence of the school, especially from teachers and principals. Based on the data assessed, the relationship with teachers is notably a positive factor in promoting physical activity. Similarly, the decisions made by the administrative team concerning the sports practiced in the school community are relevant, as are whether they make the decision to make a difference between girls and boys.

With regard to principals, the data assessed present this group as strategic for intervention purposes in terms of physical inactivity at school. This is because the measures adopted by this group will influence the experiences of many students, as well as the community as a whole. An example of the decisions made concerning the sports to be promoted in their schools is as follows:


*The problem is that this was originally a men’s school, so soccer at school has always been a male thing … for example, even during breaks … And this is the case of basketball too, but not of other (x) ... activities or sports that have been implemented later. In fact, girls began to play sports, and later the course included girls playing soccer. Track, for example, has always been considered to be more feminine, if I am not mistaken, right? And in fact, girls … always perform well, so to speak.*

*(Private School No. 3, Principals)*


This quotation from the administrative team notes that the physical activities mostly promoted by educational establishments are sports, and these, in turn, have a historical and cultural burden that takes root in school communities. In this case, the burden is associated with gender, as soccer and basketball are related to men, whereas track is more linked to women. This would affect the school community in general by offering differential possibilities to students. However, some girls began to play soccer, indicating that gender-based sports are not sensible to students.

Although several educational establishments have been making progress in the integration of women into sports, an aftertaste is still perceived regarding the male-oriented perspective, as shown in the following paragraph:


*Student No. 1: When I played basketball, when lessons were face-to-face, I remember that there were not so many balls for girls, as we use different balls. Boys for instance, use No. 7 and girls No. 6 balls, and sometimes there were no No. 6 balls for all of them, and good ones, because some were made of a material that not everyone likes. So we had to use a smaller ball, a No. 5, or use balls for boys (No. 7). That was the negative thing that often happened.*

*(Private School No. 2, Students)*


The above excerpt highlights that access to sports material provided by the educational establishment is an issue that can further affect students’ motivation to play sports and, therefore, the establishment’s physical inactivity figures.

As for the relationship with teachers, it was reported that once students, especially girls, reach sixth grade, they tend to be less involved in the physical education class, as they feel their bodies are more exposed and the menstrual cycle is crucial in that connection. At this point, it was noted that this moment of the menstrual period could be addressed with a supportive bond between the teacher and the students, as indicated in the following excerpt.


*Student No. 2: But teachers are flexible, I mean, the good thing is that our teacher is a woman, so if we tell her that we are on our period, and ask if we can sit down, she agrees, so she is more flexible with that, of course.*

*(Private School No. 2, Students)*


This flexibility from the PE teacher can be perceived as empathy toward students, as “a female teacher” can understand what students go through during their menstrual cycle. When a teacher’s gender is emphasized, students highlight that male teachers have fewer possibilities of empathizing with them, a situation that opens up the possibility for male teachers to develop pedagogical tools to understand and empathize with this situation.

Another factor to consider is the teacher’s practice, which may be mediated by their expectations with regard to what men and women can do in terms of physical demands. The data assessed accounted for the differentiation and categorization of physical exercises by gender, a practice that is not enjoyed by students, as can be seen in the following excerpt.


*Student No. 1: And physical education classes are mixed-sex. What happened to me in the other school is that we were divided into male and female groups, rather than into courses. We were separated and, if the class was small, they blended seventh and eighth grades together, for instance. Or in case of larger classes, eighth grade A and B were grouped together, but boys and girls did not exercise together. And here the case was totally the opposite, being all together. There was no differentiation between strength exercises (for boys), track (for girls) as they did in my former school.*

*Student No. 2: The difficulty of the exercise only depends on the ability of each individual, regardless the gender.*

*(Private School No. 1, Students)*


The quote shows a comparison between two types of PE classes: one that makes a distinction between boys and girls, and another that does not do so. Student No. 1 highlights that mixed-sex classes are “a totally different thing,” which is in contrast with the gender separation that she experienced in the other school. This comparison shows that both methods coexist in our time, leading to various effects on individuals. In view of this, the expression “the difficulty of the exercise only depends on the ability of each individual, regardless of gender” explicitly indicates that the differentiation in PE class should be based on the difficulty of an exercise for each individual, rather than the assumption that boys will do “strength exercises” while girls would be engaged in “gymnastics.”

The data obtained showed different aspects of facilities. In this sense, the use of educational establishment areas was significant.


*Teacher No. 1: One remark about that: I agree with Rosita, who speaks about the interest of girls at that age, but if we take into account the physical distribution of every educational establishment, and we take a look at fields, which are great areas … (Who uses the field?) Which is the group that makes use of the field?*

*Several teachers: [Boys]*

*Teacher No. 1: [boys], who play soccer, exactly, so girls stand at the sides. Have you noticed that? Girls are always at the side of the fields, so stereotypes also begin there. We are always fostering our reduction as women in sports, even in the distribution of the physical area available.*

*(Public School No. 1, Teachers)*


In this sense, a very important situation that is not so widely acknowledged is the use of facilities. Although students’ access to facilities is relevant, their use is equally significant. Data reveal that these areas are mostly used by boys who play soccer to the detriment of other potential activities and games. This situation occurs in both public and private spaces, thus suggesting that the spaces available to play, such as the backyard and break areas, are not democratically distributed between boys and girls and those who do not wish to play soccer.

### 3.4. Community Level

The community dimension refers to the services and relationships between organizations, including the safety of the neighborhood and the accessibility of facilities crucial in reducing physical inactivity.

With regard to safety, the risk level identified in a neighborhood is considered relevant when deciding to exercise in a square.


*Well, I don’t know, the people who cannot afford a … well, one says “you could go exercise in the square’s sports area,” but it depends on the area where you live, because if the square of your neighborhood is very insecure, you will not go there to play sports in the morning. I mean, there are lots of factors that fall outside the policies that this or previous administrations have been able to execute.*

*(Private School No. 3, Teachers)*


Based on the above, a teacher highlights the “economic access” of individuals, as this determines whether an individual is “playing sports in the square.” This conditioning factor is linked to the sense of security in public areas, which suggests that physical inactivity may be associated with factors that go beyond the individual dimension, something that would “fall outside the policies,” regardless of the government in power.

As for access to facilities, the possibility of playing sports or different physical activities in squares is also regarded as a relevant aspect.


*Interviewee No. 5: There should be more just open grounds, or maybe a larger amount of fitness equipment in squares, I don’t know. In most squares, there is fitness equipment or areas to play sports. Yes, maybe if we had more open grounds or stuff like that to play more sports, or maybe more clubs to be enrolled in within the municipality, in different sports, not just soccer teams.*

*(Public School No. 1, Students)*


With regard to what has been stated above, students mention their need to have more squares available, and the expression “not just soccer” highlights that they want areas that allow them to play “different sports.” This situation is relevant, especially in the public system, as access to sports and physical activities outside school is conditioned, as indicated below, by each family’s economic access. In this sense, from a social justice perspective, the public area should be one to challenge inequalities in access to physical activities, something that cannot be achieved.

### 3.5. Public Policy Level

The public policy level considers national and local initiatives in addition to laws that may have an impact on individuals’ physical inactivity. The data assessed show that standardized testing, PE curriculum, and education in general, and several cultural aspects that public policies do not critically analyze, have a role at school, influencing physical inactivity. The following excerpt delves into this:


*How ashamed girls were and, based on objective data, the other SIMCE indicators, which also measure academic self-esteem, we found that their self-esteem was really low, especially in the second cycle, so these self-esteem workshops were created (…) Self-esteem is closely related to physical activity so, as we said earlier, if girls … there are high rates of obesity at school, and they feel fat and try to hide their bodies. This is why they do not want to do physical activity, so that would be a subject to research too.*

*(Public School No. 1, Teachers)*


Another relevant aspect that the above quote makes reference to deals with the standardized SIMCE test, which reveals that self-esteem is related to doing physical activity. In specific terms, teachers build a local correlation of the results of this test, mentioning that the “self-esteem” indicator is objectively low, which would guide the decisions made by the school. They particularly associate self-esteem with obesity, indicating that because there is low self-esteem and high obesity rates, involvement in PE classes will decrease.

In relation to the above, public education establishments are those that notoriously adopt more state plans and programs, turning them into the main recipients of public policies. In this context, the fact that a centralized and standardized program developed by the Ministry of Health was adopted by an educational establishment has been raised as a problem by an administrative team, as this would lead to difficulties in the school community.


*Interviewee No. 1: Sure, this was developed by the MINSAL (Ministry of Health), and there were children who have been examined at the doctor’s office, diagnosed with this… with obesity, for instance, so it was mandatory for them to be in the program. And they felt like “I am in the program because I am fat.” So we wanted to change that view and extend our invitation for every big boy and girl who wanted to be part of the program, so that every interested person could be part of it.*

*(Public School No. 1, Principals)*


The above paragraph shows that administrative teams regulate public policies that seek to help their overweight students by implementing local criteria that aim not to differentiate the recipients chosen by the health service, as prioritized children said, “I am in this program because I am fat,” which would lead to school dynamics that may be perceived as negative to them. For this reason, they made the decision to “change that view, and extend the invitation to everyone” in order to involve the educational community and avoid promoting an individualization of neither the problem nor the solution.

Although public policies condition the actions that may be implemented by schools and individuals, the data also showed that there are given roles in society that are both targeted at men and women, and their promotion through the communication media affects physical activity rates.


*Interviewee No. 1: I also believe there is a decrease that goes beyond the school in particular, but is more related to society in general, where the role that women can play is circumscribed, delimited, previously demarcated and, therefore, they never break ground, no … unlike boys, who are more encouraged to play sports. Outstanding sportsmen portrayed by the media are generally men; women who stand out in sports or (x) highlighted by the press are always a few.*

*(Private School No. 1, Principals)*


This quote indicates that the sociocultural dimension can also play a part in physical inactivity at school, as there are roles for women that “are circumscribed, delimited, previously demarcated,” and these guidelines are linked to aspects that are not related to sports and physical activities. Therefore, if a girl wants to engage in physical activity, she will be breaking social grounds that are difficult to change. Furthermore, the fact that the media highlights sports figures who “are generally men” accounts for the lack of female role models for girls to identify with.

## 4. Discussion

The results of this analysis are in line with the WHO diagnosis regarding physical inactivity in girls and women [2,3,4,5]. In a cross-sectional manner, all educational actors and types of schools agree that girls are more inactive than boys. This situation is also consistent with the information obtained in Chile by the Ministries of Health and Sports, which suggests that girls are more inactive than boys [7,10].

Concerning the educational actors’ assessment method for physical inactivity, they also agree that it is a problem with more dimensions than just individual agency, which is in line with the criticism of the biomedical model adopted by public policies [6,16,17].

These criticisms are grounded in the evaluation of physical inactivity using the socioecological model, as there is consensus that this is a complex issue, and attempting to solve it based on individual behavioral change will not succeed [18,19]. In this sense, our results state that, at the intrapersonal level, the dimensions explicitly affecting physical inactivity are gender, self-esteem, and age. Ethnic origin was not a problem mentioned in the interviews conducted, which suggests a discrepancy with several studies [56].

As for the interpersonal level, the results are consistent with those of Hu et al.’s [56] study because friends and family are relevant factors when doing physical activity. In this regard, students make a difference between their involvement in face-to-face PE classes and those delivered via digital platforms as a result of the COVID-19 pandemic. The first modality was associated with fun and positive emotions, whereas online classes were linked to less positive feelings. Another significant aspect is the fact that, owing to the pandemic, virtual classes increased the use of electronic devices at home, which was considered a problem by families.

At the organizational level, teachers’ support has been shown to promote motivation in boys and girls in PE classes, which aligns with the study carried out by Hu et al. [51]. In this sense, it is worth mentioning that PE classes are stereotyped [26,28]. Similarly, this study’s findings are consistent with those of Castejón and Gimenez’s study, which indicated that girls prefer dancing-related activities, whereas boys choose sports [27]. Nonetheless, these decisions are conditioned by the choices made by adults (principals and teachers) without asking the students about their preferences first [23]. When students give their opinions, they state that they prefer a mixed-sex class, with no gender differences [32,34].

At this point, data encourage us to develop pedagogical tools to address the menstrual cycle in PE classes because, so far, common sense has prevailed both from the students and the professionals working at schools. This aspect would require deeper scientific research.

At the community level, it can be seen that the access to and use of the establishments’ shared areas are relevant for engaging in physical activity. For access, it is indicated that safety and the type of fitness equipment available in public squares are important for individuals. Concerning the use of space, the data show that areas are mostly used by boys in their soccer practice over girls or other activities. This can be linked to the study conducted by Hidalgo and Almonacid [71], which indicates that the main difference between boys and girls is the use of physical space for playing purposes. They also state that girls show self-marginalization attitudes, giving more prominence to boys, who tend to play soccer in common areas.

As for the last level, the results show that public policy focuses on the individual, which aligns with the criticisms made earlier [17,18,19]. In this vein, schools tend to regulate public policies for them to be consistent with their institutional identity, which is often more collective than individual.

## 5. Conclusions

In relation to the aim of this study, the socioecological model provides an alternative explanation and insight into how physical inactivity is reproduced in girls from the experiences of public, subsidized, and private schools in Chile. In this sense, it is possible to conclude that physical inactivity is conditioned by factors consistent with the socioecological model. It can be stated that women are more physically inactive than men as an effect of sociocultural factors, and that these gendered relationships with physical activity are developed in the education system. These factors are not natural and can be transformed in schools as well as from public policies. Aspects such as social construction of growth stages, gender stereotypes in PE and sports classes, access to safe squares, emotions, and public policies are aspects that can be transformed. It should be clarified that the results of this study represent public, subsidized, and private schools since the factors indicated are experienced by all the people who live in the country.

Although this research only sought to analyze how physical inactivity in girls is reproduced from school institutions, some practical implications that can collaborate in the solution of physical inactivity in girls can be identified. The first is linked to schools, understanding that management teams, teachers, and families can coordinate so that physical activity can become part of everyday life for both boys and girls. For example, workshops could be held that include children and families, or establishments could be opened after PE classes, such as on weekends. The second action is linked to policy makers. They could also adopt socioecological models to shape their work and generate actions to listen to the students and promote debate around the differences in gender that are promoted in Chilean society. Likewise, they could include in their analyses the local and territorial school cultures so that the activities are relevant in territorial terms.

To conclude, this study necessitates the continuation of scientific exploration of three areas. First, the menstrual cycle and its relationship to PE class, considering how girls indicated that they need greater confidence and openness to talk about these topics and that they feel their female teachers are more open to dialogue than male coaches. The second topic is gender separation in PE classes, as several establishments still preserve the idea that several sports are for men and others are for women. Furthermore, most students stated that they prefer mixed-sex PE classes. Finally, another issue that would be interesting to address is the effect of technology on family dynamics, as parents mentioned that due to the COVID-19 pandemic, their children were more connected to technology, thus increasing family conflicts. Policy interventions in educational settings to increase physical activity and improve public health based on an individualistic biomedical approach have failed to achieve their aims in the Chilean context; indeed, they may have contributed to even greater inequalities in physical activity based on gender. This research indicates a need to move to a socioecological model to adequately address the continuing inequality in physical activity for girls and women.

## Data Availability

Not applicable.

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
