# Peer review of "How Can Physical Inactivity in Girls Be Explained? A Socioecological Study in Public, Subsidized, and Private Schools"

_ijerph, 2022, doi:10.3390/ijerph19159304_

Round 1
Reviewer 1 Report
1. Please try to number your headings or/and make them bold, and help readers catch your points. For example, under “1. Introduction,” you have: “1.1 Physical Inactivity in Women” and “1.2 The Socioecological Model.”
2. For participants in “2. Materials and Methods,” 59 students were recruited. Did they come from elementary, middle, senior high schools or colleges? It seemed this was not clear. Different levels of education may vary greatly in physical education. Do you think this is an issue and needs to be addressed in your study?
3. Again for participants, “The sample comprised eight educational establishments: three representing the public context, three subsidized schools, and two private institutions.” Did they come from elementary, middle, senior high schools or colleges? What are the difference between subsidized schools and private schools? Usually it is possible that private schools can get some financial support from government. Additionally, readers would be confused with “context, schools, and institutions.” Did they all mean “schools” or not? If not, what are the differences among them? For international readers, it would be hard to understand this.
4. For participants, more clarification is required. For example, since you mentioned that “The sample comprised eight educational establishments,” then it would be reasonable to guess there would be “eight” principals. However, you said that you had 31 principals in your interviews. This is confusing.
5. For participants, more clarification is required. Another example, who were non-teachers (N = 37)? Why were they important to be included in the study?
6. What is the sampling method? Did participants come from the same organizations or not? Were students and families related? How could you access the participants? Why were they willing to accept the interviews?
7. For the interview outline, you mentioned “the third section sought information on the complexities observed by each role when reducing physical inactivity at school.” Can you provide some examples of questions you asked in the interviews to help readers understand the data collection? Particularly, readers would be wondering whether you had a major research target in your research. Were “female students” your major target? Or did you mean “general females”? How did you specify this in your interview outlines?
8. Since this study focused on “Physical Inactivity in ‘Girls,’” comparison between male and female perspectives would be important because females seemed to look at themselves while males seemed to look at others. However, your study did not provide this comparison. How many male and female participants were there in your study? Do you think the comparison is important or not?
9. Between page 10 and 12, you repeated the section, “organization level.”
10. It is important to clarify the difference between organization and community. In “3.3 organizational level,” you also mentioned “school community.”
11. In “3.4 Community Intrapersonal Level,” here “Intrapersonal Level” was also mentioned. Didn’t you think this would be confused with “3.1. Intrapersonal Level”?
Author Response
Dear Reviewer:
Many thanks for all your comments! We responded to you on a separate document. Please see the attachment.
Kind regards,

Reviewer 2 Report
Hello.
The article falls into both the field and addresses an interesting topic.
The research presentation is well done, orderly and explicit.
The argument and the introduction are solid and support the idea of research. The methods are clearly presented, as well as the content of the research.
Congratulations. All the best!.
Author Response
Dear Reviewer:
Many thanks for your comments.
Kind regards,
Reviewer 3 Report
The study seeks to qualitatively assess the complexities suffered by schools to reduce physical inactivity, based on the opinion of the actors affected by public policies. Considering the criticism made to the biomedical model, gender dimensions that condition women to be more inactive than men, in this study, a new approach is proposed through the implementation of the individualistic model to solve the problem of reduced physical activity.
A significant sample was selected for the study, which comprised eight educational establishments: three representing the public context, three subsidized schools, and two private institutions. This study allowed for the gender-based assessment of physical activities that different educational actors (principals, teachers, non-teachers, students, and family) promote to reduce physical inactivity at school.
As a result of the study, the authors conclude that physical inactivity is conditioned by sociocultural factors consistent with the socioecological model. Ultimately, the authors make practical recommendations for continuing of scientific exploration of three areas: the menstrual cycle and its relationship to the physical education class, gender separation in the physical education class, effect caused by technology in the family dynamics.
Despite the high relevance of the study, the practical significance for solving the problems of motor activity is not completely clear.
The purpose of the study was not specifically set and it is not clear how it was eventually achieved. The differentiation of various types of educational institutions stated in the topic was not reflected in the conclusions, discussions and conclusions. The abstract presented before the main text of the article does not reflect the main essence of the study.
Overall Recommendation: Reconsider after major revision.
Author Response
Dear Reviewer:
Many thanks for your comments. Please see the attachment with our responses.
Kind regards,

Round 2
Reviewer 3 Report
The study seeks to qualitatively assess the complexities suffered by schools to reduce physical inactivity, based on the opinion of the actors affected by public policies. Considering the criticism made to the biomedical model, gender dimensions that condition women to be more inactive than men, in this study, a new approach is proposed through the implementation of the individualistic model to solve the problem of reduced physical activity.
A significant sample was selected for the study, which comprised eight educational establishments: three representing the public context, three subsidized schools, and two private institutions. This study allowed for the gender-based assessment of physical activities that different educational actors (principals, teachers, non-teachers, students, and family) promote to reduce physical inactivity at school.
As a result of the study, the authors conclude that physical inactivity is conditioned by sociocultural factors consistent with the socioecological model. Ultimately, the authors make practical recommendations for continuing of scientific exploration of three areas: the menstrual cycle and its relationship to the physical education class, gender separation in the physical education class, effect caused by technology in the family dynamics.
Overall Recommendation: Accept in present form.